# Plant-Derived Exosomes as a Drug-Delivery Approach for the Treatment of Inflammatory Bowel Disease and Colitis-Associated Cancer

**DOI:** 10.3390/pharmaceutics14040822

**Published:** 2022-04-08

**Authors:** Ying Cai, Luoxin Zhang, Youjian Zhang, Rong Lu

**Affiliations:** 1Marine College, Shandong University, No. 180 Wenhua West Road, Weihai 264209, China; 201900810250@mail.sdu.edu.cn (Y.C.); 201900810242@mail.sdu.edu.cn (L.Z.); 202000810025@mail.sdu.edu.cn (Y.Z.); 2Newland Biotechnology Co., Ltd., No. 213 Huoju Road, Weihai 264200, China

**Keywords:** plant-derived exosomes, nanoparticle, drug delivery, anti-inflammatory, inflammatory bowel disease, colitis-associated cancer

## Abstract

Inflammatory bowel disease (IBD) is a chronic recurrent intestinal disease and includes Crohn’s disease (CD) and ulcerative colitis (UC). Due to the complex etiology of colitis, the current treatments of IBD are quite limited and are mainly concentrated on the remission of the disease. In addition, the side effects of conventional drugs on the body cannot be ignored. IBD also has a certain relationship with colitis-associated cancer (CAC), and inflammatory cells can produce a large number of tumor-promoting cytokines to promote tumor progression. In recent years, exosomes from plants have been found to have the ability to load drugs to target the intestine and have great potential for the treatment of intestinal diseases. This plant-derived exosome-targeting delivery system can load chemical or nucleic acid drugs and deliver them to intestinal inflammatory sites stably and efficiently. This review summarizes the pathophysiological characteristics of IBD and CAC as well as the application and prospect of plant exosomes in the treatment of IBD and CAC.

## 1. Introduction

IBD is a chronic recurrent intestinal disease, with a rising incidence in the last few decades, which has contributed to complex and dysfunctional immune responses [1]. Patients’ intestinal microbial imbalance, intestinal barrier damage, or susceptible genes may cause intestinal inflammation [2]. Moreover, up to 20% of IBD patients are more likely to suffer from CAC, and the incidence increases with the prolongation of the disease [3]. Current conventional therapies for IBD are mostly oral chemical drugs, including 5-aminosalicylic acid derivatives, corticosteroids, immunosuppressants, and biotherapy using anti-necrosis factor drugs. Patients often need lifelong maintenance drugs, because colitis is difficult to cure [4]. However, long-term use of these conventional drugs often affects the human immune system and leads to obvious side effects. Therefore, a new drug delivery system (DDS) is needed to improve the therapeutic effect of drugs in intestinal disease [5,6]. Current delivery methods for therapeutic drugs include the synthetic DDS (such as liposomes and microspheres) and natural exosomes (such as plant-derived exosomes and mammalian-derived exosomes). Unfortunately, the synthetic nanoparticles can lead to cell stress, apoptosis, activation of inflammatory bodies, and other side effects [7,8]. Experiments on rats have shown that synthetic nanocarriers can enhance immune reactivity, stimulate the expression of Hsp70-2a and Hsp90, and cause serious DNA or RNA damage [7]. Exosomes derived from mammalian cells have great prospects as therapeutic vectors; however, the separation and utilization of these exosomes creates challenges for mammalian exosomes to deliver multiple molecules to target cells [9,10]. In addition, they are difficult to obtain in large quantities and can activate the host’s immune response. Plant-derived exosomes may help overcome these challenges [11,12]. Plant-derived exosomes are a kind of membrane vesicle with a nanometer size [13], released from edible plants such as ginger, lemon, strawberry, blueberry, grapefruit, tea, and so on [14,15,16]. With the progress of the research, the components of exosomes from plants have gradually become clear, including lipids, proteins, and nucleic acids, and sometimes contain special chemical components related to the plants that are plant-specific [17,18,19]. Researchers have found and reported the existence of exosome-like nanoparticles in more and more plants, fruits, and mushrooms [20]. Exosomes from edible plants, such as grapefruit, tomato, blueberry, and shiitake mushroom, have anti-inflammatory function [21]; apple [22] and carrot exosomes may affect intestinal transporters [23]; citrus and lemon exosomes have antioxidant properties and antitumor effects [24,25,26]. Plant-derived exosomes have excellent biochemical characteristics, and their inherent biological functions are as follows [27,28]:(1)Good biocompatibility;(2)No toxicity and low immunogenicity;(3)Specifically targeting ability;(4)Extending the cycle period and prolonging the action time of the drugs;(5)Production capability in large quantities;(6)Crossing the Blood–Brain Barrier (BBB) but not the placenta.

The specificity of plant-derived exosomes given by specific orientations and their ability to manipulate genes for treatment, transfer hydrophobic drugs, and escape immune attacks make them suitable for DDS as future medical applications [29]. When faced with IBD and CAC, the drug targeting system with plant-derived exosomes as the carrier provides a new method for treatment, which may realize the targeted delivery of inflammatory medicines [30,31,32]. Studies have shown that a variety of chemicals and nucleic acid drugs can be effectively delivered by plant-derived exosomes to intestinal inflammatory sites, thereby reducing inflammation or inhibiting gene expression [33]. This new drug delivery method has been proven to have a good therapeutic effect on intestinal inflammation [34]. In this paper, we review and clarify the possibility and clinical application prospect of plant exosomes as a new generation of drug delivery nanoplatforms for the treatment of IBD and related cancers. More importantly, we summarize the different types of drugs delivered by plant exosomes and their therapeutic effects, aiming to highlight new ideas for the treatment of IBD and CAC in the future.

## 2. Pathogenesis of IBD and CAC

IBD is a chronic recurrent intestinal disease with a rising incidence in past decades. CD and UC are two main forms of IBD, characterized by intestinal injury and massive inflammation. The two diseases have similar pathological and physiological manifestations, such as acute pain, vomiting, and diarrhea; however, there are differences in the pathogenesis, which also distinguish CD and UC from two different diseases and may require different treatment methods. Long-term IBD may lead to complications, and the risk of colon-related cancer is greatly increased [35,36]. At present, the precise etiology of IBD is still unclear [37]. Here, we review the effects of intestinal microflora, susceptible genes, and the intestinal microbial barrier on IBD (see Figure 1). The most important hypothesis suggests that IBD is caused by an excessive immune response induced by the changes of intestinal microflora or pathogenic microorganisms in susceptible hosts.

### 2.1. Gut Microbiota

The most important hypothesis suggests that IBD is caused by excessive immune response, induced by environmental factors, to the changes in intestinal microflora or pathogenic microorganisms in susceptible hosts [38]. Given that the dysregulated immune response to gut microbiota is considered to be a fundamental pillar of the development of chronic intestinal inflammation, the complex interactions between the host and gut microbiota must be precisely regulated.

Human intestinal microflora is a dynamic and diverse community with multiple bacteria [39]. Under normal physiological conditions, intestinal microflora is involved in complex physiological and biochemical processes as a dynamic equilibrium organ. In previous studies, significant changes in gut microbiota have been confirmed between IBD patients and healthy people [40]. At the same time, the composition and state of intestinal microflora are also quite different between CD and UC. In the intestinal flora of CD patients, the range of inflammation is widely distributed in the intestine, and the dominant factor may be the invasion of harmful bacteria. However, UC is an acute mucosal disease usually confined to the distal colon, and the loss of beneficial bacteria may overwhelm the invasive bacteria. According to previous research, gut microbiota-based therapies have shown good results in improving colitis in animal models and IBD patients, suggesting that gut microbiota may be a promising therapeutic target for IBD. In addition, the plant-derived exosomes can be selectively absorbed by intestinal flora, regulating the expression of some genes and the intestinal microenvironment [41,42].

### 2.2. Genetic Factors

Through the study of adult gut microbiota, genetic factors have been proven to affect the composition and structure of gut microbiota. Host genotypes have a lasting effect on the composition of gut microbiota. For example, the composition of gut microbiota in monozygotic twins shows high similarity, and individuals living in the same environment and having similar eating habits show the least similarity [43]. Compared with the general population, relatives of CD patients have a much higher risk of CD. Studies have shown that siblings of CD patients have a relative risk of CD more than 30 times higher than the general population [44]. The inheritance of IBD seems to be related to hundreds of common genetic variations. Genome-wide association studies (GWASs) have identified about 200 different IBD susceptibility loci that affected the production of cytokines and the activation of intestinal immune cells (such as macrophages and neutrophils) [34]. For example, *TNFSF15* can be expressed in a variety of tissues, including myeloid cells, activated T cells, and endothelial cells. As a susceptible gene, *TNFSF15* can promote immune pathology by signaling its receptor DR3 and then stimulate cytokine expression to increase the risk of IBD [45]. In addition, *C1orf106* is also a susceptible gene for IBD, which causes intestinal epithelial barrier defects and increases susceptibility to intestinal pathogens [46]. Unfortunately, only a few IBD susceptible loci have been functionally characterized so far. More studies are needed to determine the mechanisms, and we look forward to more researchers joining this field in the future.

### 2.3. Intestinal Microbial Barrier

The intestinal epithelium itself is covered by the mucus layer, forming the first layer of physical defense [47]. Mammalian intestinal mucus is composed of two systems. The loose outer layer of mucus is composed of mucin and diluted antimicrobial agents, which are the normal habitat of symbiotic bacteria, while the inner layer is firmly attached to the epithelial cells; it is rich in antimicrobial agents and shows low bacterial density. The physiological correlation and protective function of the mucus layer have been proven, and the lack of mucus layer can lead to colon inflammation and put bacteria directly into contact with the intestinal epithelium [48]. In IBD, the defense and tolerance mechanisms of intestinal microorganisms are impaired at multiple levels, resulting in biological disorders and associated changes in the intestinal environment, thereby further weakening the intestinal epithelial barrier and increasing epithelial permeability [49]. Epithelial barrier dysfunction is followed by increased translocation to lamina propria. In lamina propria, defects in antigen treatment may cause some strong inflammatory responses.

More and more evidence shows that the interactions among gut microbiota, barrier, and epithelial and mucosal immune cells show a complex dynamic network under strict regulation. In IBD, the defense and tolerance mechanisms of intestinal microorganisms are impaired at multiple levels, resulting in biological disorders and associated changes in the intestinal environment and leading to further weakening of the intestinal epithelial barrier and increasing epithelial permeability [50].

## 3. Existing Nanocarriers and Limitations

In current studies, a variety of nanocarriers have been invented to load drugs for treating intestinal inflammation [51,52]. Due to their unique size and size-dependent physical properties, nanoparticles can move stably in the gastrointestinal tract and pass through the mucus layer to intestinal cells (Figure 2)[53]. Inflammation of intestinal epithelial barrier defects can increase the permeability of the intestinal barrier, thereby promoting the penetration of nanoparticles (NPs). Additionally, inflammatory mucosa increases mucus secretion, which promotes the adhesion and diffusion of NPs through the intestinal mucus layer [54]. Gene silencing caused by RNA is a candidate treatment for intestinal inflammation [11]. siRNA is a powerful tool for post-transcriptional silent gene expression, which can interfere with the expression of specific genes [55]. Liposomes have the potential for targeted drug delivery [56,57,58]. However, as a DDS, what limits the development of liposomes is their quality; their purity and stability still need to be improved.

Exosomes derived from mammals are also a DDS with great potential, having the advantages of low immunogenicity and high tissue targeting, and a large number of experiments have been carried out using these. For example, curcumin encapsulated by exosomes can improve stability and anti-inflammatory properties [59], and exosomes derived from brain endothelial cell culture can improve the penetration of doxorubicin and PTX on BBB [60]. In addition, exosomes from tumor cells can serve as drug carriers for efficient delivery of anticancer drugs [61]. Unfortunately, challenges remain in expanding production and isolation based on current exosome separation technologies, especially given the size overlap between exosomes and other extracellular vesicles. Moreover, mammal-derived exosomes can be quickly removed from the blood circulation after in vivo administration, which is not conducive to drug retention in the body [62].

In addition to liposomes, many other delivery systems have been developed [54]. For example, some natural products represented by pectin and chitosan have been proven to have drug-loading capacity and can prevent the premature release of drugs in the intestine [63]. In addition, there are some synthetic nanoparticles (such as polymer micelles, polymersomes, and inorganic nanoparticles). However, the synthesized nanoparticles have many disadvantages compared to plant-derived exosomes. First of all, their side effects on the human body are uncertain, while the toxicity of plant-derived exosomes is low. Second, the biocompatibility of synthetic nanoparticles is poorer than that of plant sources [64,65,66]. At the same time, when plant sources are internalized, they can induce stem cell proliferation, activate internal and external apoptotic pathways, and inhibit tumor growth and progression, which are advantages that synthetic nanoparticles do not have [28].

These nanoparticle-based oral administration systems have significantly promoted the prospects for the treatment of colon-related diseases, mainly by improving the selective targeting of therapeutic agents to inflammatory sites, reducing the systemic toxicity of drugs, and providing possibilities for the encapsulation of new compounds with relatively unstable properties [67]. As far as current research progress is concerned, although the delivery system shows high stability, these parameters need to be validated in vivo using mature disease models [68,69]. From the perspective of commercial development, we need to simplify drug delivery design to achieve efficient and reliable mass production.

## 4. Plant-Derived Exosomes Have the Ability to Treat IBD and CAC

### 4.1. Isolation and Purification of Plant-Derived Exosomes

The separation and purification of exosomes are mainly achieved by the combination of differential ultracentrifugation and sucrose density gradient centrifugation techniques [70,71]. Ultracentrifugation is inadequate in securing extra cleansed exosomes; this can be achieved by gradient ultracentrifugation, although an additional processing period is required [72]. The extraction and purification technology of plant exosomes is constantly improving. Liu et al. included a 0.22 μm filtration step that preferentially separated vesicles of sizes that may be the most suitable for therapeutic delivery, and its applicability in cruciferous plants was tested [73]. Leaf exosomes are basically complete and have a uniform and appropriate size and surface charge, which may be compatible with therapeutic delivery. In addition, an efficient method based on electrophoresis and dialysis was proposed, which was time-saving and needed no special equipment [74,75,76]. Other methods, such as ultrafiltration and immune separation, have been used in animal-derived exosomes to obtain more pure preparations, but they have the disadvantage of high cost and are not widely used in the purification of plant-derived nanovesicles (PDNVs). The immunoaffinity method also has high potential, but it has not been widely studied for the separation of plant-derived exosomes, which may be due to the lack of extensive understanding of plant-derived exosome surface composition and antibody-antigen interaction [77].

### 4.2. Characterization and Constituents of Plant-Derived Exosomes

The plant exosomes obtained by the above methods were characterized. The plant exosomes from different sources had certain characteristics in terms of particle size, Zeta potential, and other aspects. The normal size of exosomes ranges from 30 to 1000 nm [73]. However, there are some differences in the size of exosomes from different sources, such as ginger and grapefruit exosomes with an average size of 250 nm, and coconut-derived exosomes with particle size of less than 100 nm [71]. In general, plant-derived exosomes showed a negative ζ potential, ranging from −100 mV to 0 mV, indicating that they are mutually exclusive and lack aggregation tendency [23]. Plant exosomes can remain stable at different temperatures and pH values (different plant sources may show some differences). Exosomes have a unique multi-layer flower-like structure, which can better load proteins, nucleic acids, chemical drugs, etc. [78].

Plant extracts may be able to offer the original therapeutic effect of plants but cannot be directly used as DDS. Compared with plant extracts, plant-derived exosomes have carrier properties and targeting properties and can extend the cycle period. Natural exosomes from plants are full of bioactive lipids, proteins, ribonucleic acids, and other pharmacologically active molecules, varying from their mammalian exosome counterparts [79,80]. Lipid is a key component of the exosome structure, which is mainly divided into two types, phospholipids and glyceryl, without cholesterol [34,81]. Protein is an integral part of the function of exosomes, roughly divided into transmembrane proteins and other plasma membrane-related proteins [82,83]. In addition, the nucleic acids in exosomes are mainly RNA, including mRNA, microRNA(miRNA), and small RNA (sRNA) [84,85,86]. After entering the recipient cells, mRNA and non-coding miRNA can regulate the levels of RNA and protein in the cells, affecting gene expression and cell morphology and function [87,88] (Figure 3) [89]. In addition, plant exosomes also concentrate important active plant derivatives [21]. For example, exosomes from broccoli are rich in sulforaphane, which is known to be an active compound extracted from this vegetable. Similarly, ginger exosomes contain 6-gingerol and 6-shogaol, which are two active ingredients of ginger with anti-inflammatory properties [90].

### 4.3. Plant-Derived Exosomes Can Relieve Intestinal Inflammation

According to the recent literature, plant-derived exosomes display many biological properties, which are illustrated in [70]. At the same time, experiments have proven that exosomes from plants have regulatory effects on a variety of anti-inflammatory cytokines in the intestine. Taking oral ginger-derived nanoparticles (GDNPs) as an example, recent research shows that there are 27 miRNAs with high expression levels in GDNPs, which are mainly involved in the regulation of inflammation and cancer-related pathways. In addition, GDNPs can resist lipopolysaccharide (LPS)-induced inflammation by down-regulating the expression of NF-κB, IL-6, IL-8, and TNF-α, indicating that GDNPs have the potential to reduce damage factors while promoting healing [91,92]. Different plant exosomes also have differences, for example, carrots tend to only induce IL-10 [23]. This proves that plant exosomes can also remain stable in the gastrointestinal environment to ensure the stable delivery of drugs to the target site. Through in vitro and in vivo toxicity studies, exosomes derived from plants are usually non-toxic and non-immunogenic and cannot pass through the placental barrier, which indicates that exosomes are safe as DDS [93].

Besides being absorbed by intestinal cells, plant exosomes can also be absorbed by intestinal microflora to regulate the intestinal microenvironment and alleviate inflammation [88]. Recent studies have found that lemon-derived exosomes enhance the tolerance of lactobacillus to bile, which plays a key role in shaping intestinal flora; this proved that lemon-derived exosomes could increase the stress survival rate of intestinal bacteria [94]. The exosomes from edible mulberry bark can prevent colitis by the AhR/COPS8 pathway, accelerating the healing of intestinal mucosa [32]. In addition, plant exosomes have colon targeting, which means they can be taken up by macrophages and intestinal stem cells, which can effectively remain in the intestine, pass through the intestinal mucosal barrier, ensure the targeted delivery of drugs to the intestine, and improve the local concentration of drugs [95]. According to the research by Wang et al., the anti-inflammatory drug methotrexate (MTX) can be added to grapefruit-derived nanovesicles (GDNs). The results suggest that GNDs may act as an intestinal immunomodulator to maintain intestinal macrophage homeostasis. Compared with free MTX, the transfer of MTX-GDN to mice could significantly reduce the toxicity of MTX and significantly improve its therapeutic effect on DSS-induced colitis in mice [12].

## 5. Colon Targeting Mechanism of Plant-Derived Exosomes

### 5.1. Electrostatic Interaction

Charged particles can accurately target tissues with opposite charges under certain conditions, so electrostatic interactions can be used to target areas of intestinal inflammation. Due to the presence of phosphate, the surface of plant-derived exosomes is usually negatively charged, and particles are dispersed because of the repulsion of the same charge [79]. In the inflammatory region, there are some substances like negatively charged carbohydrate moieties; however, positive proteins such as transferrin and eosinophil cationic proteins are abundant [96]. Negatively charged plant-derived exosomes preferentially adhere to the inflammatory region by electrostatic interactions with these proteins. In a rat colitis model, the adhesion of liposomes with different charges to inflammatory regions was compared in vitro. The results showed that the accumulation of negatively charged liposomes in inflammatory tissue was twice as high as that of neutral or positively charged liposomes [97]. In addition, when the electrostatic interaction with the mucus layer is weak, negatively charged NPs are more likely to penetrate the mucus layer than positively charged NPs. In conclusion, these results provide evidence that NPs with different charges can interact with gastrointestinal components through electrostatic interaction, and in theory, they can give target specificity for diseased sites. At the same time, the experiment proved that plant-derived exosomes also had a certain targeting effect on macrophages, which could induce the expression of HO-1 and IL-10 in macrophages and enhance the anti-inflammatory effect of macrophages. In addition, the production of proinflammatory cytokine TNF-α was significantly inhibited [12].

### 5.2. Ligand-Modified NPs

Ligand-modified NPs targeting inflammatory regions can increase drug accumulation in inflammatory bowel tissues and reduce drug redistribution in normal tissues [97]. Studies have shown that hyaluronic acid (HA) can specifically bind to CD44 overexpressed in inflammatory epithelial cells and macrophages of colitis [98]. In another study, HA was modified by polymerized NPs to deliver tripeptides to colitis tissue [99]. The directional ability of plant-derived exosomes can be actively accelerated by providing selective peptides or ligands on the surface of exosomes, which may target recipient cells holding related receptors. The folic acid receptor has been confirmed to be overexpressed in inflammatory response. Therefore, plant-derived exosomes using FA as a ligand can effectively utilize the levels of FA receptors in various tumor cells, and an enhanced homing effect was found in mice transplanted with CT26 or SW620 cells, thereby reducing tumor volume [72].

The using of monoclonal antibodies (mABs) is an important biological therapy for IBD. Monoclonal antibodies against pro-inflammatory cytokines (such as TNF-α) target the crosstalk between IL-23, IL-6, IL-17, and TGF-β. Monoclonal antibodies against adhesion molecules (such as ICAM-1) can inhibit the recruitment of effector T cells [54]. The combination of antibodies and peptides to increase drug targeting is also a potential method. However, due to the effects of enzymes in the body circulation and physiological conditions in the gastrointestinal tract, multiple factors must be considered in application [97]. In addition, antibodies are usually associated with the risk of inducing immune response.

## 6. Plant-Derived Exosomes as Drug Carriers for the Treatment of IBD and CAC

In the new carrier system introduced in recent years, exosomes from plants as a DDS has great potential [100,101]. Compared with conventional methods, plant-derived exosome delivery has the advantages of improving drug absorption rate, targeting delivery to intestinal inflammatory sites, high biocompatibility, and non-toxic side effects. As nanocarriers of drugs, they have unique morphology and composition characteristics, which may be able to meet the increasingly stringent requirements of current clinical challenges [102]. Plant-derived exosomes are also less toxic than synthetic lipid nanoparticles. When injected intravenously into pregnant mice, they do not pass through the placental barrier, indicating that they may be a useful drug delivery tool [103]. This DDS formed by exosomes extracted from plants has great potential for treating IBD and CAC. [97].

Usually, a large proportion of cytotoxic drugs used by patients do not reach the tumor but are distributed throughout the body, leading to the many toxic effects associated with chemotherapy and thereby reducing their therapeutic effects [104,105] In contrast, the exosomes are derived from edible plant tissues, which are composed of biocompatible and biodegradable materials [77]. They have the ability to encapsulate multiple drugs and can be attached to specific types or groups of cells in a targeted manner [106,107]. They protect the therapeutic agents from degradation and directly transport them to the diseased site [108]. Plant exosomes also have excellent drug loading capacity, which can efficiently load chemicals, nucleic acids, and peptides. In the experiment of Zhang et al., where DOX was loaded with ginger-derived exosomes, DOX had a slightly positive charge, and most ginger lipid components had a negative charge. With the help of ultrasound, ginger lipids could form a bilayer, and DOX could be loaded into ginger-derived exosomes by electrostatic interaction. When the ginger lipid concentration was 0.1 mol/l, the loading efficiency was as high as 95.9 ± 0.26% [98,109]. Mao et al. delivered anti-TNF-α antibodies to the colon via an oral route, showing high efficacy in colitis mice and reducing severe systemic adverse reactions during intravenous injection [110]. Plant exosomes can load different drugs, and their biological effects in vivo will not change, which is an important aspect of improving the transport of hydrophobic drugs in particular [111,112]. Although exosome loading and release capacities may vary for different types of reagents, existing experiments have shown that this is a very efficient, stable, and safe delivery method.

### 6.1. Chemical Medicine Loading by Plant-Derived Exosomes

For improving the therapeutic effect of drugs on IBD or CAC and reducing the systemic toxicity of free drugs to normal tissues and organs, loading drugs into plant exosomes has been proven to be an effective method [113] (Table 1). For example, grapefruit-derived nanovesicles (GDNs) were coupled with immunosuppressants and anti-inflammatory methotrexate (MTX), and then the drug was delivered by oral administration. The extracellular microenvironment of tumor cells was acidic (pH 6.5~6.9), and the release performance of Dox-GDNVs was tested in a biological model. The experimental results showed that Dox-GDNVs could stably release DOX when the pH value was about 6.5, which indicated that Dox-GDNVs could release drug delivery faster in acidic pH close to the tumor microenvironment [109]. In the comparison of cell anti-proliferation experiments between methotrexate carried by GDNs (GMTX) and MTX, GMTX showed a similar inhibitory effect on cell proliferation to MTX, through flow cytometry analysis. Subsequently, the observation of cell targeting proved that GMTX successfully targeted most MTX to macrophages in the lamina propria. The preparation was used for the treatment of DSS-induced colitis in mice. The comparative experiments showed that the GMTX treatment group had a better effect on slowing the colon tissue damage and reducing the range of colitis cells in mice than the free MTX groups. It effectively reduced specific inflammatory cytokines (including tumor necrosis factor-α and interleukin-1β at mRNA and protein levels) and effectively reduced DSS-induced weight loss and colon length shortening [48,114]. This experiment proved that exosomes from some plants do not adversely affect the intestine, but enhance the anti-inflammatory ability of the host, providing strong evidence for the development of oral drug delivery carriers based on plant exosomes [115,116].

Furthermore, grapefruit-derived nanovectors (GNVs) can deliver STAT3 inhibitor JSI-124 to tumor sites through non-invasive pathways, significantly inhibiting STAT3 activation. Wang et al. proved that GNV encapsulated with STAT3 inhibitor JSI-124 (12.5 pmol/10 mL) could effectively inhibit tumor growth. In addition, GNVs can be modified to achieve better tumor targeting, such as high-affinity folate receptors (FRs) expressed in many human tumor cells, and the expression of FRs in non-tumor cells can be negligible [98]. The two tumor transplantation models, mouse CT26 colon cancer model and human SW6 colon cancer SCID mouse model, showed that oral administration of GNVs-FA could enhance the GNVs-FA signal in the stomach and small intestine of mice, indicating that the GNVs combined with FA could enhance the retention of GNV in the intestine [122]. The longer the retention time was, the greater the possibility of penetrating tumor tissue. In addition, the experiment proved that the accumulation in the spleen was significantly reduced after the surface binding of FA, and the accumulation in the liver was slightly reduced, which proved that the use of this preparation could reduce the systemic toxicity of drugs to normal tissues or organs [123]. The combination of FA and chemotherapeutic drugs through GNVs can efficiently deliver the chemotherapeutic agents to the tumor site. In the experiment, the tumor growth was significantly reduced and the tumor volume was significantly smaller than that of other drugs. Similarly, Zhang et al. combined ginger-derived nanovectors (GDNVs) and FA, then evaluated the in vivo targeting and anti-tumor ability of Dox-FA-GDNVs, and studied the biological distribution of Dox-FA-GDNVs and FA-GDNVs in vivo [91]. Under the condition of establishing DiR-labeled FA-GDNVs into mice bearing Colon-26 tumors, the anti-tumor ability of Dox-FA-GDNVs was explored. The experimental results showed that Dox-FA-GDNVs significantly inhibited tumor growth, markedly reduced tumor weight, and significantly promoted tumor tissue cell apoptosis, reaching 43.3%. These results showed that GDNVs with FA ligand could further enhance the anti-tumor effect of free DOX. In summary, FA combined with plant-derived exosomes can be used as an efficient drug-targeting delivery platform, which can effectively improve the therapeutic effect of drugs, prolong the circulation time of drugs in the blood, reduce the systemic toxicity of free drugs, and reduce side effects.

### 6.2. Nucleic Acid Loading by Plant-Derived Exosomes

There are two main strategies for miRNA-centered treatment. One is that miRNA inhibits the synthesis of single-stranded RNA, which leads to the up-regulation of specific proteins by inhibiting the effect of endogenous miRNA. The other is that miRNA enhances the synthesis of miRNA, which is used to simulate endogenous miRNAs, to ensure the same effect by prohibiting the degradation of mRNA. MiRNAs and small interfering RNAs (siRNAs) are the two main categories of small RNAs, which are processed by single-stranded hairpin RNA precursor or double-stranded RNA (dsRNA), with a length of about 21–24 nucleotides [124]. For the gut microbiota, a given miRNA may have hundreds of different bacterial mRNA targets, and hundreds of miRNAs may be encapsulated in an exosome simultaneously to achieve efficient drug delivery [125]. In recent years, siRNA drugs have been considered to have great potential for the treatment of colitis and related cancers [126,127]. However, due to the poor absorptive capacity and poor targeting ability of cells to siRNA, they are easily eliminated by the kidney system and often need to be encapsulated into nanoparticles for delivery [117]. The currently used synthetic nanoparticles have potential side effects and non-specific limitations. Therefore, it is necessary to develop a new siRNA DDS to enhance the accumulation of drugs in tumor tissues, maximize the therapeutic effect, and minimize the adverse reactions [128,129].

Several experiments have shown that exosomes derived from plants are effective carriers of siRNAs, with good biocompatibility and able to be produced economically on a large scale. Thus, they have great potential for drug delivery and may change the current siRNA delivery mode (Figure 4). CD98 is a type II transmembrane protein. Overexpression on the surface of colon epithelial cells and macrophages promotes the occurrence and development of IBD [130]. Therefore, it is reasonable to speculate that targeted blocking of CD98 expression may provide an effective way for the treatment of IBD.

In the experiment of Zhang et al., siRNA-CD98 was encapsulated by ginger-derived exosomes, and the encapsulation efficiency was as high as 61 ± 8%, which proved that ginger-derived exosomes could be used as efficient carriers of siRNA. When siRNAs are encapsulated in exosomes, their biological effects remain unchanged in vivo. siRNA-CD98/GDLV was delivered orally. The results showed that siRNA-CD98/GDLV maintained high retention in the colon and ileum, while the free DIR as the control was only retained in the stomach, which effectively proved the high targeting of siRNA-CD98/GDLV to the intestine. In addition, siRNA-CD98/GDLV specifically reduced the expression of the CD98 gene in the mouse colon, weakened the inflammatory response, and effectively alleviated colitis and colitis-related cancer.

Similar to previous studies, exosomes bound with FA can also deliver siRNA efficiently, which significantly improves the delivery efficiency of siRNA. Wang et al. proved through experiments that the injection of GNVS-FA-siRNA-Luc could reduce the expression level of luciferase in CT26 tumor cells by more than five times under the same conditions. This is undoubtedly an effective way to improve the delivery of RNA drugs such as siRNAs/miRNAs. However, some other nucleic acid drugs have been proven to have good effects in the treatment of colitis, such as antisense oligonucleotide against TNF-α. Huang et al. loaded them into synthetic nano-preparations and achieved good anti-inflammatory effects by oral administration [118]. Therefore, perhaps these nucleic acids can also be loaded into plant exosomes for better results.

## 7. Conclusions and Discussion

Plant exosomes are naturally filled with bioactive lipids, proteins, ribonucleic acids, and other pharmacologically active molecules as natural nanocarriers, which have unique morphological and composition characteristics. They can not only accurately package different tumor necrosis factors, but also package fluorophores, targeting agents, small molecule drugs, proteins, or other therapeutic drugs. Compared with synthetic nanoparticles, naturally derived exosomes are expected to be produced economically and have the possibility of large-scale production. Since the pathogenesis of IBD is still unclear, more experiments need to be carried out to further explore the effects of intestinal microflora and the immune mechanism of the host itself on inflammation. Due to the fact that the efficient regulation of gene expression by siRNA can minimize the toxicity of intestinal inflammation, we believe that this targeted delivery of siRNA therapy in the future will draw more and more attention. In addition, compared with exosomes loaded with drugs or siRNA alone, the co-delivery of drugs and siRNA by exosomes may have more advantages in cancer treatment, which may also become a new idea for drug delivery. At the same time, surface modification of exosomes has been proven to be beneficial to improve their targeting ability and stability. However, harmful electrostatic interactions between charged exosomes and other charged components of the gastrointestinal tract (such as bile acids and soluble mucins) are still possible. In order to fully build the system of targeted drug delivery by plant exosomes and find more efficient and economic nanocarriers, more plants need to be studied.

Nowadays, plant-derived exosomes as a therapeutic carrier have entered a stage of rapid development, and some of them have entered clinical studies. However, before they become a mature DDS for clinical use, there are still many serious challenges. More research is needed on the selection of suitable plant sources and purification technologies for mass production. In the near future, more and more scientists can be expected to participate in this new field.

## Figures and Tables

**Figure 1 pharmaceutics-14-00822-f001:**
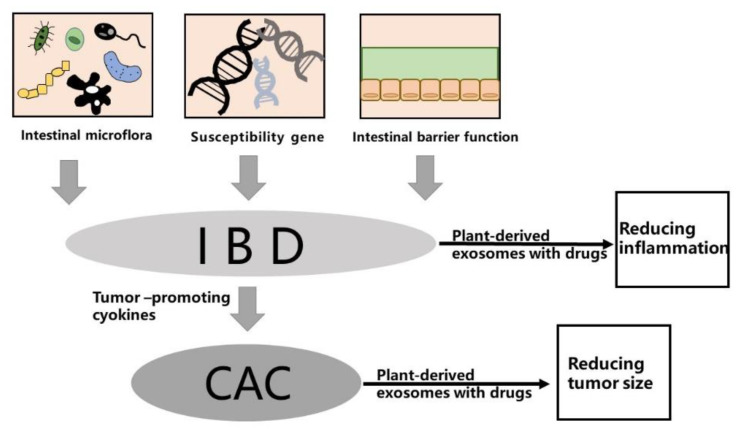
Typical factors for IBD. There are many factors for IBD, among which intestinal microflora, susceptibility gene, and intestinal barrier function are widely recognized. Furthermore, inflammatory intestinal epithelial cells can secrete cytokines that promote tumor growth. Experiments show that IBD and CAC can be effectively treated by plant-derived exosome-loaded drugs.

**Figure 2 pharmaceutics-14-00822-f002:**
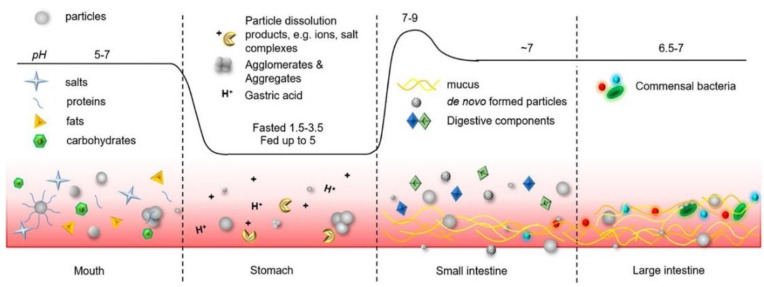
Oral administration can lead the nanoparticles to intestinal epithelial cells and load therapeutic drugs to enter epithelial cells, which are not affected by other substances in the gastrointestinal tract.

**Figure 3 pharmaceutics-14-00822-f003:**
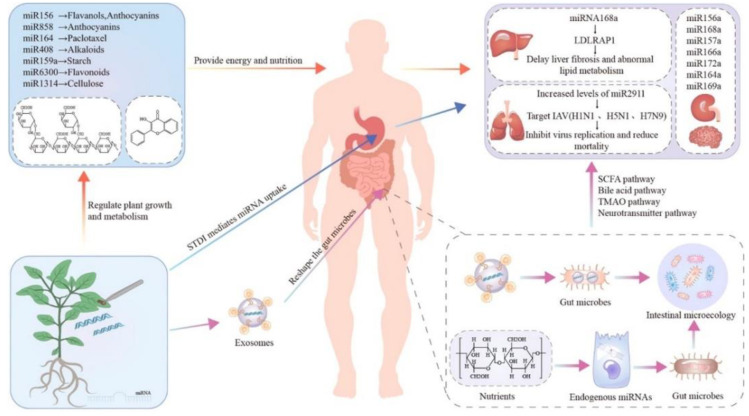
The intake of plant miRNA may have many effects on the body. For example, some plant miRNAs target host cell mRNA after absorption in vivo, inhibit the post-transcriptional splicing or translation process of target mRNA, and affect protein expression. At the same time, plant miRNA can enter the intestine and affect the host intestinal flora, thereby interfering with metabolism and physiology.

**Figure 4 pharmaceutics-14-00822-f004:**
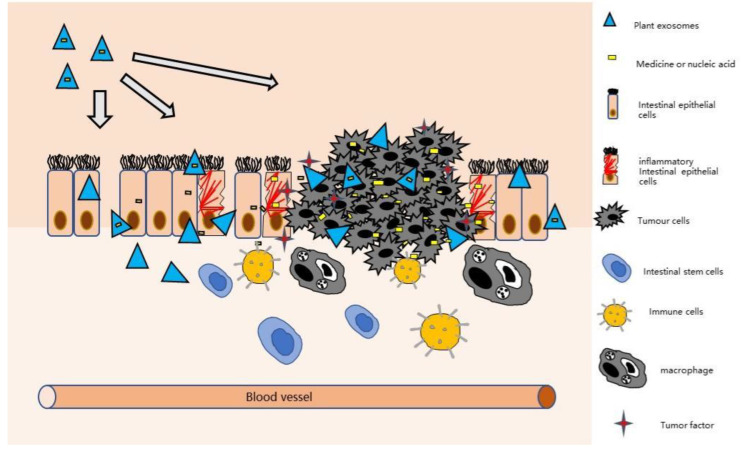
Plant-derived exosomes loading drugs to inflammatory intestinal epithelial. The exosomes secreted by plants can load chemical drugs or nucleic acid drugs to effectively treat CAC and reduce the size of tumors. At the same time, plant-derived exosomes can affect intestinal epithelial cells, macrophages, and other intestinal cells to reduce intestinal inflammation and create a more healthy intestinal microenvironment.

**Table 1 pharmaceutics-14-00822-t001:** Current delivery systems for disease treatment.

Drug	Delivery System	Target	Disease	Reference
17-AAG	PLGA	Heat shock protein 90(Hsp90)	UC/CAC	[3]
6-gingerol/ 6-shogaol	Ginger-derived exosomes	TNF-α/IL-6/IL-1β/IL-10/IL-12	IBD	[91]
	PLGA/PLA-PEG-FA			[103]
MTX	Grapefruit-derived exosomes	Macrophage	IBD	[88]
Dox	Ginger-derived exosomes/ Ginger-derived exosomes combined with FA	Cellular DNA	CAC	[109]
siRNA-CD98	Ginger-derived exosomes	Epithelial/Macrophage	UC	[117]
JSI-124	Grapefruit-derived exosomes/Grapefruit-derived exosomes combined with FA	Stat3	Brain tumor	[70]
Dtxl	PLGA	Microtubule	Tumor	[105]
CPT	Synthetic nanoparticles	Epithelial	CAC	[113]
Cy3-siRNA	Termed thioketal nanoparticles (TKNs)	TNF-α	IBD	[118]
ASO	Amphoteric liposomes	CD40	IBD	[58]
Indocyanine green (ICG)	Aloe-derived exosomes	Melanoma cell	Melanoma	[119]
MiR-21 inhibitor and chemotherapeutics	Engineered exosomes	Cellular DNA	CAC	[120]
RAGE-binding peptide (RBP)	Engineered exosomes	Cytokine assays in macrophage cells	Lung inflammation	[121]

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
