# Peer review of "Plant-Derived Exosomes as a Drug-Delivery Approach for the Treatment of Inflammatory Bowel Disease and Colitis-Associated Cancer"

_pharmaceutics, 2022, doi:10.3390/pharmaceutics14040822_

Round 1
Reviewer 1 Report
"Changes have been made according to my suggestions. I recommend the acceptance.
Regards,"
Author Response
We thank you for the time and effort that you have put into reviewing the manuscript. And we apprecaite so much for your affirmation of our work.
Reviewer 2 Report
I had the opportunity to read and review the previous version of the manuscript and I acknnowledge the good job made by the authors to clarify and address my main concerns. I think this investigation will be of interest to researchers interested in exosomes or natural drug-delivery approaches.
But I still detect a few issues on how the ideas and words expressed on the manuscript are expressed.
For instance:
Lines 24-26: "IBD is a chronic recurrent intestinal disease with a rising incidence since the last few decades,which has contributed to complex and dysfunctional immune responses"
Lines 28-29: "Moreover, up to 20 % of IBD patients are likely to suffer from CAC, and the incidence increases with the prolongation of the disease".
I´d recommend a more careful proofread, before being accepted for publication.
Reviewer 3 Report
The title of the review manuscript is very interesting but the running text has several shortcomings that are listed as follow
- Authors need to cite some recent work on plant-derived exosomes with experimental explanation and figures
- The review length and contents are too much brief, specially part 2.2 and 5.2
- Why on earth authors have deleted figures 2 and 3 in the revised manuscript?
- Authors need to read these articles to make better their review
Plant-derived exosome-like nanoparticles and their therapeutic activities, Asian Journal of Pharmaceutical Sciences, 2022, 17(1), 53-69.
Plant-derived exosome-like nanoparticles: A concise review on its extraction methods, content, bioactivities, and potential as functional food ingredient, Concise Reviews & Hypothesis in Food Science, 2021, DOI: 10.1111/1750-3841.15787
Tumor exosome-based nanoparticles are efficient drug carriers for chemotherapy, Nature Communication, 2019, 10: 3838.
Round 2
Reviewer 3 Report
The language should be improved before being published
This manuscript is a resubmission of an earlier submission. The following is a list of the peer review reports and author responses from that submission.
Round 1
Reviewer 1 Report
Overall, while this investigation could be of interest to researchers interested in exosomes or natural drug-delivery approaches. I cannot recommend this version of the manuscript for publication in the present form. It is poorly written and unclear in certain sections. I´d recommend a more careful proofread, before being accepted for publication.
- The objective of a review paper should be to become a critical reference to what has been published in the field (Lines 63-65). This manuscript should include a more critical approach and lacks references in important sections. For instance, in the introduction section (Lines 34-37) the authors describe how synthetic nanoparticle might lead to “cell stress, apoptosis and activation of inflammatory bodies” but do not describe the mechanisms on how synthetic nanoparticles induce those effects and do not include any references to support the statement. In fact, I think this should be a section on the manuscript itself. Comparing the properties of natural vs. synthetic nanocarriers.
- Nanoscience is a young field, and it is rapidly evolving. As general rule it has been accepted that nanotechnology applies to engineered and natural materials that have at least one dimension in the size range 1 – 100 nm. But nanoparticles are a heterogeneous mixture of particles of different sizes, shapes, chemical composition, crystal, and amorphous structures and with coatings and surface chemistry. In particular, coatings and surface chemistry can, to some extent, control their behavior and, more importantly, often change properties depending on their surrounding media. It is surprising that the authors based their arguments to colon target mechanisms only in terms of size and zeta potential to explain the interaction of exosomes in terms of size and electrostatic interactions (Lines 231-238). While there are so many other particle-particle and particle-surface interactions which may be helpful.
- Some typos are found in the manuscript (e.g., “PH” line 186; “m77acrophages” line 253).
- Some acronyms are found in the manuscript but haven´t been described (e.g., “PDVN” line 177, “ELD” line 174).
- It is recommended that if authors cite other authors work and refer to a name, they include the citation after the name (e.g., Liu et al. line 169; Wang et al. line 216).
- In section 3 “Existing nanocarriers and limitations”, it is necessary to review more bibliography on the field. The authors assert in the manuscript: “In addition to liposomes, many other delivery systems have been developed,” but these are not described. I´d recommend restructuring this section.
- Either section 7 is missing or there is a mistake in the numbering.
Reviewer 2 Report
The article is an interesting compilation of studies carried out with plant-derived exosomes applied to Inflammatory Bowel Disease and Colitis-associated Cancer but some additions to the content would improve the manuscript.
Minor changes: figure captions should be changed since they include information usually reserved for the text. Captions should be only a brief description of the figure. Any additional information, even if supported and explained in the figure, should be explained in the text itself, not in the captions.
Major changes:
If the therapeutic activity associated with exosomes is mainly due to their content, what is the significant improvement assignable to plant-derived exosomes over traditional plant extracts? There are only glimpses in the text; the authors should clarify this point explicitly.
Finally, comparison with similar applications but based on mammalian-derived exosomes could be further explored, as these are the most studied vesicles. The authors could compile the most significant studies on the same pathologies and molecules loaded in mammalian cell-derived exosomes and highlight the differential facts offered by plant-derived exosomes compared to those of animal origin.
Reviewer 3 Report
The title of the review manuscript is very interesting and important to conduct this study but the running text has several shortcomings that are listed as follow
- The author did not discuss the extraction methods in detail, types of plants, and the proteins, Table 1 showed limited data, however, there are many reports on this topic. The conclusion and discussion part are also in the initial stage. The authors failed to explain the future prospects of the study. The relationship between plant drived exosomes and pathway is missing. According to the above serious concerns, I think authors need to provide a comprehensive study about this topic. The present form of the manuscript is infancy.
- Authors need to give an insight into microflora, and microbiota variants.
- Authors need to avoid the use of abbreviations in the keyword section (Page no.1, line no. 20)
- What kind of immune cells have a role in the regulation or activation? (Page no.4, line no. 120)
- No information has been provided about genes (specific name) and their function in IBD (Section 2.2, Page no.3-4, line no. 107-120)
- Authors need to provide a pictorial demonstration of Section 2.3 on Page no 4.
- Whether use RNA interference or gene silencing on Page no. 4, line no. 144
- Kindly improve the structure of the sentence on Page no. 4, line no. 147-148
- Is it a typing error or the name of the cell?? on Page no. 8, line no. 253
- Kindly write unite according to the guideline of SI Unite (Page no 8. line no. 288).
- Authors need to revise the sentence on Page no. 8, line no. 285 and on Page no. 10 line no. 337. They seem repeated.
- Kindly italicize the word in vivo and in vitro throughout the manuscript